# From the Sun to the Cell: Examining Obesity through the Lens of Vitamin D and Inflammation

**DOI:** 10.3390/metabo14010004

**Published:** 2023-12-20

**Authors:** Alina Delia Popa, Otilia Niță, Lavinia Caba, Andreea Gherasim, Mariana Graur, Laura Mihalache, Lidia Iuliana Arhire

**Affiliations:** 1Faculty of Medicine, University of Medicine and Pharmacy “Grigore T. Popa”, 700115 Iasi, Romania; alina.popa@umfiasi.ro (A.D.P.); andreea.gherasim@umfiasi.ro (A.G.); laura.mihalache@umfiasi.ro (L.M.); lidia.graur@umfiasi.ro (L.I.A.); 2Faculty of Medicine and Biological Sciences, University “Ștefan cel Mare” of Suceava, 720229 Suceava, Romania; graur.mariana@gmail.com

**Keywords:** adipose tissue, obesity, senescence, vitamin D, inflammation

## Abstract

Obesity affects more than one billion people worldwide and often leads to cardiometabolic chronic comorbidities. It induces senescence-related alterations in adipose tissue, and senescence is closely linked to obesity. Fully elucidating the pathways through which vitamin D exerts anti-inflammatory effects may improve our understanding of local adipose tissue inflammation and the pathogenesis of metabolic disorders. In this narrative review, we compiled and analyzed the literature from diverse academic sources, focusing on recent developments to provide a comprehensive overview of the effect of vitamin D on inflammation associated with obesity and senescence. The article reveals that the activation of the NF-κB (nuclear factor kappa B subunit 1) and NLRP3 inflammasome (nucleotide-binding domain, leucine-rich-containing, pyrin domain-containing-3) pathways through the toll-like receptors, which increases oxidative stress and cytokine release, is a common mechanism underlying inflammation associated with obesity and senescence, and it discusses the potential beneficial effect of vitamin D in alleviating the development of subclinical inflammation. Investigating the main target cells and pathways of vitamin D action in adipose tissue could help uncover complex mechanisms of obesity and cellular senescence. This review summarizes significant findings related to opportunities for improving metabolic health.

## 1. Introduction

Adipose tissue (AT), traditionally considered an inert organ primarily responsible for energy storage and release, has emerged as a subject of increasing research interest in recent decades. Studies have revealed that white adipose tissue (WAT) has far more intricate functions than previously realized, exhibiting numerous connections and being responsible for multiple processes within the body. As one of the largest endocrine organs, WAT actively synthesizes hormones such as adiponectin, adipsin, apelin, leptin, resistin, and visfatin. These hormones are involved in regulating inflammation, insulin secretion, insulin sensitivity, and food intake, and they respond to both environmental and internal cues by releasing proinflammatory cytokines [1].

In addition to adipose cells, the stromal vascular fraction (SVF), which is regarded as the local “immune system”, plays a crucial role in AT. It surrounds crown-like structures and perivascular spaces and comprises mesenchymal, endothelial, and immune cells such as macrophages, eosinophils, and Th_2_ CD4+ T cells [1]. These cells interact with sympathetic nerve endings and produce neurotrophic factors, impacting local sympathetic activity [2]. Diverse macrophage subpopulations in AT can determine whether the environment is pro- or anti-inflammatory, with M1 macrophages releasing proinflammatory cytokines (TNFα, IL-1β, IL-12, and IL-23) and M2 macrophages producing anti-inflammatory cytokines (IL-10) [3,4]. White-adipose-tissue-resident multipotent stromal cells (WAT-MSCs) have been shown to support both group-2 innate lymphoid cell (ILC2) activity and the proliferation of adipose tissue eosinophils (ATEs); the latter play a crucial role in maintaining the M2 polarization of macrophages [5,6].

In humans and rodent models, obesity is characterized by an imbalance in the growth and size of adipocytes, with AT exhibiting hypertrophy in comparison to the surrounding vascular tissue [7]. This imbalance results in regional hypoxia, apoptosis, chemokine release, and inflammatory cell recruitment. This process initiates local inflammation, later leading to systemic subclinical inflammation [7]. The metabolic disruptions contribute to the ongoing cycle of AT dysfunction and inflammation, resulting in the development of a diversity of systemic problems in patients with obesity.

Ageing leads to a rise in the percentage of body fat and a shift in the distribution of AT from the subcutaneous layers to visceral layers (visceral AT, VAT). AT senescence increases the accumulation of senescent cells and alters the preadipocyte cell phenotype, resulting in elevated secretion of proinflammatory cytokines, which consequently contribute to the inflammageing of VAT.

Although inflammation is common in both obesity and senescence, there are key differences between obesity-related AT inflammation and inflammageing with respect to their causes and mechanism and the types of cells and inflammatory cytokines involved. Obesity is associated with early and pronounced AT senescence, and AT senescence increases the risk for obesity, suggesting that obesity and senescence, while not identical, interact and contribute to the development of subclinical inflammation associated with insulin resistance and metabolic syndrome.

Individuals with both obesity and vitamin D insufficiency exhibit elevated levels of proinflammatory cytokines, such as IL-6 and TNFα (tumor necrosis factor α). An umbrella meta-analysis showed beneficial results for vitamin D supplementation on obesity-associated inflammation [8]. Moreover, treatment with vitamin D suppresses the generation of inflammatory markers induced by the NF-κB (nuclear factor kappa B subunit 1) pathway in human adipocytes and preadipocytes. Furthermore, the molecular mechanism by which vitamin D supplementation impacts NF-κB signaling in hypertrophic and inflamed AT is not fully understood [9]. Of particular significance is the phenomenon of adipose cell senescence, and its link to obesity has recently attracted significant research attention. The NF-κB pathway is just one of the common pathways through which vitamin D may modulate both local inflammation and inflammageing. Identifying the common target cells and pathways of vitamin D in adipose cells and the SVF may be of interest for better understanding the mechanisms of obesity and AT senescence. This avenue of investigation has the potential to yield valuable insights and targets for improving metabolic health. By exploring the process of adipose cell senescence and its association with obesity, researchers may uncover the key mechanisms involved in the development of cardiometabolic disorders. The significance of these findings lies in their potential implications for improving the metabolic outcomes of treatment strategies for obesity.

## 2. Materials and Methods

We initially searched the following databases and libraries to obtain many relevant studies: PubMed, EBSCO, and Embase. We focused on peer-reviewed articles published in English and excluded conference abstracts and editorials (Table 1). Reviews, randomized controlled trials (RCTs), and observational studies published up until October 2023 were included in the analysis.

## 3. Results

### 3.1. Adipocyte Dynamics in Obesity: Connections with Cellular Senescence

During the ageing process, damage from various agents accumulates in cells, which triggers senescence, leading to the loss of the cells’ ability to divide and exhibit specific characteristics [10]. Cellular senescence occurs due to various factors: DNA damage, telomere shortening, endoplasmic reticulum (ER) stress, mitochondrial dysfunction, mitotic stress, oxidative stress, and oncogene activation [11,12,13]. DNA damage, especially DNA double-strand breaks, activates a response pathway, culminating in the activation of the p53/p21 axis, thereby inducing cell cycle arrest [13]. This is characterized by irreversible cell cycle arrest in the G1 or G2 phase, orchestrated by tumor suppressor proteins and kinases such as p53, p16, and p21 [1,14,15]. Although senescent cells do not proliferate, they remain metabolically active, displaying alterations in gene expression and chromatin structure [1].

AT is particularly susceptible to ageing, as it is among the most-affected tissues by age-related deterioration [16] and is one of the organs where cellular senescence begins earliest [17]. Life expectancy declines with obesity, and fat mass increases with age in both mice and humans. Caloric restriction increases lifespan, which is attributed to a reduction in the volume of VAT depots [18]. Obesity is associated with age-related changes and an elevated number of senescent cells in AT [19].

Cellular senescence, while advantageous for tissue repair and tumor suppression, can be detrimental when chronic, such as in obesity. One of the primary regulators of lipid homeostasis, SREBP1c (sterol regulatory element-binding protein 1), known to modulate cholesterol and fatty acid metabolism, appears to play a crucial role in protecting against adipocyte senescence. Obesity-induced DNA damage triggers adipocyte senescence. The protective role of SREBP1c involves the maintenance of genome stability, increasing PARP1 (poly (ADP-ribose) polymerase 1)’s DNA repair capability. Adipocytes deficient in SREBP1c show increased inflammation, implying that adipocyte senescence might be an early event preceding inflammatory responses in WAT in the context of obesity [20,21].

Senescence and obesity both result in increased cellular dysfunction and inflammation. However, senescence is characterized by an increase in the number of nondividing, inflammatory cells, while obesity is characterized by expansion and proliferation of adipocytes, which are accompanied by metabolic disruptions (Figure 1).

#### 3.1.1. Senescence-Associated Secretory Phenotype (SASP)

A notable characteristic of the senescent cells is their unique secretome, known as the senescence-associated secretory phenotype (SASP). This includes elevated senescence-associated β-galactosidase (SA β-gal) activity, an upregulated expression of cyclin-dependent kinase inhibitor-1 A (CDKNlA) and p53, and an increased production of proinflammatory cytokines and chemokines [22,23]. SASP includes inflammatory and profibrotic factors, which can induce secondary senescence in adjacent cells in a manner modulated by transcription factors, such as NF-κB and C/EBP (CCAAT/enhancer-binding protein) [10]. SA β-gal activity increases in AT as the body mass index (BMI) increases [24]. In patients with obesity, the number of senescent SA β-gal-positive cells is significantly higher in visceral white adipose tissue (vWAT) than in subcutaneous tissue (scWAT). Furthermore, people with normal weight were found to have a lower abundance of SA β-gal-positive cells in vWAT [25].

#### 3.1.2. Preadipocyte Changes in Obesity and Senescent AT

The shift in AT distribution from subcutaneous to visceral depots associated with ageing [1,26] results from the impaired differentiation of mesenchymal progenitors into adipocyte-like cells [2,26].

A series of transcription factors orchestrate adipogenesis and preadipocyte differentiation from pluripotent stem cells under the guidance of specific secretory proteins. Although the preadipocyte count remains stable or even increases with age, there are significant alterations in preadipocyte functionality, characterized by decreased replication and differentiation, increased susceptibility to lipotoxicity, and elevated proinflammatory cytokine production [24]. As these cells age, they secrete markers of the SASP, including activin A, an antiadipogenic factor [27]. The concentration of activin A is strongly associated with age and the occurrence of metabolic syndrome, making it a reliable biomarker for senescence [28]. Activin A plays a crucial role in the process of adipogenesis, as its autocrine suppression increases the differentiation of precursor cells into adipocytes [29]. Furthermore, the level of activin A in adipocyte progenitors is elevated in patients with obesity due to factors secreted by macrophages. Moreover, the concentration of activin A is higher in obese individuals than in those with a normal weight [29].

Elevated activity of SASP markers, particularly p53 and p16, is observed in senescent preadipocytes with impaired differentiation [27]. Ageing and lipotoxicity exacerbate the transformation of preadipocytes into macrophage-like entities, which promotes their proinflammatory effects [1,19].

The number of senescent preadipocytes increases in individuals with obesity, with the number of these cells being up to 30 times higher in people with obesity than in lean individuals [19]. SA β-gal-positive cells are more numerous in cultured preadipocytes obtained from obese animals than in those isolated from lean controls [24]. In individuals with obesity, there is a notable increase in SA β-gal expression in preadipocytes within AT. Studies suggest that these senescent cells, coupled with their SASP, could be the driving forces behind increased immune cell infiltration into AT. Evidence supporting this theory comes from experiments showing that when 3T3-L1 preadipocytes are cocultured with macrophages, preadipocyte TNF-α secretion increases. This implies a reinforcing loop between preadipocytes and macrophages, which exacerbates AT inflammation [1].

#### 3.1.3. Neural and Vascular Dysfunction in Obesity and Senescent AT

Ageing disrupts neuron-immune interactions [2], which is related to diminished sympathetic stimulation in elderly individuals. This affects lipid turnover, promotes WAT accumulation, reduces starvation resistance, and potentially regulates body temperature [30]. The exact mechanisms underlying reduced adipocyte lipolysis in ageing remain unclear. Despite normal catecholamine signaling, the diminished adipocyte lipolysis associated with ageing is believed to be governed by AT macrophages through an NLRP3 (NLR family pyrin-domain-containing 3) inflammasome-dependent mechanism [1,31]. Studies in mice and humans have shown increased norepinephrine degradation, which affects lipolysis, highlighting the significance of norepinephrine catabolism in ageing-associated changes in WAT [31].

Additionally, vasculature dysfunction manifests as microvascular rarefaction due to altered vascular endothelial growth factor (VEGF) signaling in aged mice (Figure 1). However, VEGF-treated mice show an increased lifespan and decreased age-related dysfunction, such as visceral obesity and inflammageing [32]. During physiological ageing, VAT releases profibrotic factors, notably transforming growth factor beta (TGFβ) and osteopontin (OPN). OPN plays roles in cell–matrix interactions across various cell types. Adipose myofibroblast differentiation is influenced by pathways such as the hypoxia/HIF1α (hypoxia-inducible factor 1-alpha) and TGFβ pathways. Platelet-derived growth factor receptor alpha (PDGFRα^+^) precursor cells, related to beige adipocytes, can become fibrogenic, leading to adipose fibrosis [33].

Endothelial cells in AT play a pivotal role in lipid transportation and maintaining WAT stability. Obesity disrupts this equilibrium, causes vascular damage in AT, and induces inflammation [1,34]. VAT-derived microvascular endothelial cells (MVECs) from obese individuals exhibit a more pronounced senescent phenotype than subcutaneous AT-derived MVECs [35]. A study on mice revealed that high-fat diet consumption not only precipitates obesity but also provokes distinct and noteworthy alterations in the gene expression profile of endothelial cells. Furthermore, these transcriptional changes are related to a spectrum of detrimental effects, such as an increased expression of p53 in endothelial cells. Cell senescence decreases the activity of PPARγ (peroxisome proliferator-activated receptor γ) in endothelial cells, impacting their response to fatty acids and lipid transport [4].

In individuals with obesity, oxygen supply to AT is insufficient as a result of AT expansion without a proportional increase in the number of blood vessels. This leads to the development of hypoxic areas, triggering the activation of HIF1α and the recruitment of proinflammatory cytokines [1,36].

#### 3.1.4. Low-Grade Inflammation in Obesity and Inflammageing

Ageing is associated with an increase in low-grade inflammation and the dysregulation of the immune function (inflammageing) (Table 2) [37]. Immune cells with proinflammatory properties (CD8+ T cells, Th1, Th17, Tγδ, and B cells) infiltrate the SVF and increase the abundance and size of crown-like structures of AT in animals and humans with obesity [34].

Ageing induces alterations in macrophage activity, accompanied by disrupted phagocytosis and cellular metabolism. The macrophage ratio shifts towards the proinflammatory phenotype, with a decrease in M2 macrophages and CD11c CD206 double-negative macrophages. The levels of proinflammatory cytokines (IL-6, IL-1β, and TNF-α) produced by M1 macrophages are elevated in the AT of older individuals [2]. NLRP3 inflammasome activation in older macrophages elevates proinflammatory cytokine release, contributing to age-related fat accumulation in the bone marrow, impacting hematopoiesis and bone health [1,2,3,40,41]. In addition, aged adipocytes have higher levels of ceramide, which activates the NF-κB pathway and induces the proinflammatory cytokines [42].

In obese animals, the first inflammatory cells recruited to AT are neutrophils, followed by monocytes/macrophages and then lymphocytes B, T, and NK cells [36]. Moreover, M2 macrophages acquire the proinflammatory M1 phenotype [34]. An important finding is that the percentage of adipose tissue macrophages (ATMs) in the visceral fat of humans with obesity increases from 4% to 12%. Furthermore, weight gain leads to the recruitment and clustering of a population of ATMs that exhibit a high expression of M1 genes but lack galactose N-acetyl-galactosamine-specific lectin 1 (MGL1). The aforementioned cells aggregate near necrotic adipocytes, which exhibit increased expression of CD11c and toll-like receptor 4 (TLR4), as well as elevated synthesis of inflammatory markers (Table 2) [3].

With ageing, ATEs exhibit shifts in distribution and function. Eotaxin, a pro-ageing factor, accumulates with age, and an increase in its expression is associated with a change in ATE distribution, leading to inflammageing and immunosenescence. Young eosinophils restore WAT function in aged rodents to reduce inflammation, emphasizing eosinophils’ role in AT homeostasis [2,43,44,45].

The number of proinflammatory T cells, such as CD8+ T cells and CD4+ (Th1/Th17), increases with ageing in AT [1]. In aged mice, regulatory T cells (Tregs) maintain AT homeostasis, emphasizing the importance of immune modulation in ageing [7,46]. In animal studies, high-fat diet consumption was found to decrease the levels of the anti-inflammatory cytokine IL-10 in Tregs from VAT. The development, maintenance, and specific gene expression patterns of VAT-Treg cells depend on IL-33 [45,47]. Moreover, Tregs express ST2, the IL-33 receptor, which is important for their accumulation and expansion in AT [48].

VAT in obese mice is rich in senescent T cells, which are uniquely positive for CD4. Upon stimulation, these cells secrete OPN, a cytokine that has been associated with inflammatory processes in the VAT of individuals diagnosed with obesity and T2DM. Furthermore, patients with obesity and hyperinsulinemia display increased expression of senescence markers in mature adipocytes. As these adipocytes become senescent, they increase in size and release higher levels of inflammatory factors (Table 2) [1,38].

In ageing mice, metabolic dysfunction in AT and mitochondrial reactive oxygen species trigger NLRP3 inflammasome activation. This disrupts B cell function through the elevation of proinflammatory cytokine levels, driving the migration of B inflammatory cells into VAT [1,31,36,49]. Moreover, the number of double-negative B cells, a proinflammatory subset, is elevated in the AT of older individuals with obesity. These metabolically active cells release proinflammatory factors and undergo metabolic changes supporting autoantibody production. The prevalence of these changes in the AT of aged individuals remains unknown [1,34,36].

#### 3.1.5. Extracellular Vesicles and Particles (EVPs)

AT secretes EVPs that transport microRNAs (miRs) resistant to degradation, which are important for interorgan communication. EVPs impact metabolic processes and obesity, indicating that AT is a key exosomal source of miRs [50]. In obesity, the expression of miRs (such as miR-27b-3p, miR-122, and miR-192) changes, leading to metabolic imbalances such as insulin resistance [50]. In the context of obesity, the activation of M1 macrophages is stimulated by the JAK-STAT (Janus kinase-signal transducer and activator of transcription) pathway in conjunction with NF-κB and MAPK (mitogen-activated protein kinase), and miR-155 promotes this process by inhibiting the production of essential anti-inflammatory mediators [51].

EVPs are also linked to cellular ageing, as they release pro-senescence molecules. Furthermore, miRs are essential players in the progression of cellular senescence; the association of miR-155 with telomere shortening underscores their role in ageing [51,52]. Ageing results in an increase in the levels of proinflammatory markers, a phenomenon suppressed by SIRT1, whose levels are modulated by miR-204. Thus, miRs play a crucial role in the complexity and regulation of senescence and may have a significant impact on the relationship between AT senescence and obesity [51,52].

Hubal et al. [53] studied the miRs profiles of adipocyte-derived EVPs from obese African-American women before and after bariatric surgery. The miR-15b-5p expression decreased significantly post-surgery, and this change was accompanied by a significant reduction in BMI. This underscores the role of adipocyte-derived EVPs in ageing and the link between AT dysfunction, ageing, and related diseases [50].

### 3.2. Physiological Roles of Vitamin D: Bone Health, Immune Function, and Beyond

In addition to its critical role in maintaining bone health and regulating phospho-calcium balance, vitamin D exerts an extensive array of effects and is a fundamental regulator of numerous physiological processes. Recent research has revealed that vitamin D regulates the expression of genes associated with the immune system, thereby exerting anti-inflammatory effects. The ability of immune cells (macrophages, B and T cells, and dendritic cells) to produce and express vitamin D receptor (VDR) provides support for this hypothesis [10].

Vitamin D acts through two main mechanisms (Figure 2). First, it acts on its nuclear receptor, VDR, which is found in several tissues. Upon binding to VDR, 1,25(OH)_2_D_3_ acts as a transcription factor that regulates the expression of certain genes. These genes can be classified as primary and secondary targets. The primary target genes include transcription factors whose expression is directly influenced by activated VDR following 1,25(OH)_2_D_3_ binding. These primary target genes subsequently downregulate the expression of secondary target genes, allowing vitamin D to exert anti-inflammatory effects [10]. Both adipocytes and immune cells express the VDR gene and enzymes associated with the effect of vitamin D. This highlights the critical role of vitamin D in ensuring the proper functioning of AT. VDR knockout mice exhibit reduced AT mass, decreased leptin levels, and increased food intake compared to wild-type mice [54]. Mice that lack *Cyp27b1* (cytochrome P450, family 27, subfamily B polypeptide 1), and thus are unable to produce 1,25(OH)_2_D_3_, show similar effects, underscoring vitamin D’s role in lipid storage [55]. The expression of genes linked with lipolysis and energy utilization is diminished in human VDR transgenic mice [56]. Vitamin D supplementation increases fatty acid oxidation [57]. Dietary vitamin D supplementation in mice following high-fat diet feeding results in a decrease in weight gain and AT mass. Increased energy expenditure and lipid utilization are responsible for this effect [58]. Vitamin D supplementation upregulates genes such as *Ppara*, *Ppargc1a*, *Ppargc1b*, *Cpt1b*, *Acadm*, and *Acadl*, which are involved in mitochondrial biogenesis and fatty acid oxidation [58].

Another mechanism through which vitamin D acts involves the assembly of a heterodimer with the retinoid X receptor (RXR). Vitamin D interacts with a specific DNA sequence known as the vitamin D response element (VDRE) in the promoter region of target genes to induce genomic changes. Moreover, vitamin D plays a crucial role in the regulation of epigenetics, largely by influencing chromatin accessibility via VDR [59,60].

Vitamin D also acts through a nongenetic mechanism that is independent of VDR and does not have an impact on gene transcription. Vitamin D binds to the membrane-associated vitamin D receptor, leading to the activation of membrane signaling cascades involving phospholipase C (PLC), diacylglycerol/inositol (DAG/IP3), protein kinase C (PKC), and MAPK, which are critical for translating extracellular signals into appropriate cellular responses [60]. Adipocyte survival, lipid metabolism regulation, insulin response, cell differentiation, inflammatory mediation, and adipokine secretion are all significantly influenced by these signaling cascades. This mechanism is essential for the preservation of cellular function and metabolic homeostasis. These pathways play an essential part in regulating fat storage and breakdown, in addition to influencing the insulin sensitivity of cells, as well as impacting the transition from preadipocytes to mature adipocytes and their involvement in inflammation and overall systemic metabolism.

#### 3.2.1. Vitamin D Deficiency in the Context of Obesity

The optimal level of vitamin D, reflected by the concentration of 25-hydroxyvitamin D (25(OH)D) in the serum, ranges from 30 to 50 ng/mL (75 and 125 nmol/L) [61,62,63]. The Endocrine Society Clinical Practice Guideline defines vitamin D sufficiency as a serum 25(OH)D level of 30–100 ng/mL [64,65]. Vitamin D insufficiency is defined as serum 25(OH)D levels between 20 and 30 ng/mL (50 and 75 nmol/L), and vitamin D deficiency is defined as a serum 25(OH)D level below 20 ng/mL (50 nmol/L) [61].

Vitamin D deficiency is a frequent finding in overweight or obese people. Initially, it was hypothesized that AT functions as the primary reservoir for vitamin D [66]. The trapping hypothesis proposes that fat cells “capture” vitamin D, preventing it from being efficiently metabolized or used by the body. This finding may explain the relationship between obesity and reduced levels of circulating vitamin D, as well as the need for individuals with obesity to take higher dosages of vitamin D to attain comparable blood levels to those with lower adiposity [11,17,24,26,27].

Individuals with obesity have a greater overall level of vitamin D in AT compared to persons with normal weight. However, it was found that the concentrations of vitamin D per gram of fat tissue were the same between individuals with obesity and those with normal weight [67]. Research suggests that vitamin D deficiency in patients with obesity may be explained by the volume dilution hypothesis, as proposed by Drincic et al. [68]. The authors contend that although the accumulation of vitamin D in both obese and nonobese individuals is comparable, the higher body weight and fat mass in patients with obesity result in a higher proportion of sequestered vitamin D [66,67].

Various systematic reviews, including those by Pannu et al. [69] and Mallard et al. [70], have indicated that weight loss can lead to a mild increase in 25(OH)D levels due to the release of vitamin D from fat stores. Concurrently, meta-analyses by Renzaho et al. [71] and Saneei et al. [72] revealed varied relationships among the levels of 25(OH)D, adiposity, and BMI. These studies emphasize that characteristics such as ethnicity, gender, and age may influence these interactions. Additionally, Vimaleswaran et al. [73] emphasized the impact of geographical region and gender on the connection between BMI and 25(OH)D levels. These findings underscore the intricate relationship among body composition, inflammation, and vitamin D, but conclusions about these associations across different populations and research methodologies remain unclear.

Research findings have suggested that persons with obesity appear to exhibit reduced 25(OH)D serum levels compared to those with healthy weight [74]; however, studies examining the association between 25(OH)D levels and fat mass have produced conflicting results [75]. Furthermore, several RCTs examining the impact of vitamin D_3_ administration on percentage fat mass (PFM) have yielded inconsistent findings. According to a recent meta-analysis, there was an observed negative correlation between 25(OH)D levels and PFM. Moreover, age, baseline BMI, and latitude were not sources of heterogeneity. However, vitamin D_3_ supplementation did not demonstrate any significant impact on PFM [76].

#### 3.2.2. The Role of Vitamin D in the Senescence Process

The results of epidemiological surveys in which vitamin D deficiency was found to be associated with a higher mortality rate suggested a role for vitamin D in the process of ageing [77]. Additionally, according to previous research, women with vitamin D deficiency have a reduction in telomere length, which is associated with a comparatively decreased lifespan. A previous study found a significant relationship between serum vitamin D levels and telomere length. This significant correlation persisted even after controlling for factors such as age, physical activity, the use of hormone replacement therapy, and menopausal status. The difference in leukocyte telomere age between subjects in the extreme tertiles of vitamin D concentrations was found to be approximately 5.0 years. This difference was further amplified by increased levels of C-reactive protein (CRP) [78]. Furthermore, mice with VDR deletion display characteristics associated with ageing [79,80]. Moreover, the administration of 1,25(OH)_2_D_3_ has been shown to elevate the expression of Klotho protein, which protects against age-related ailments such as atherosclerosis, osteopenia, and skin atrophy [81]. The involvement of vitamin D in AT senescence was suggested by Chen et al. [82]. Dietary supplementation with 1,25(OH)_2_D_3_ combined with calcium/phosphate and the antioxidant N-acetyl-l-cysteine was found to prolong the average lifespan of 1α(OH)-ase^−/−^ mice. This was due to the anti-ageing effect of 1,25(OH)_2_D_3_, which up-regulated nuclear factor erythroid 2 related factor 2 (Nrf2) and reduced reactive oxygen species (ROS) levels, DNA damage, the number of cells with the SASP, and p16/Rb and p53/p21 signaling [82].

### 3.3. Exploring the Shared Characteristics of Obesity and Ageing with a Focus on Vitamin D

Obesity is known to induce a persistent, mild, yet substantial inflammatory response, which has profound implications for overall health, particularly in the context of cardiovascular disease, diabetes, and metabolic syndrome. Cellular senescence is distinguished by the production and release of signaling molecules, including cytokines and chemotactic agents, which collectively reflect the SASP. This process is governed by many factors [52], which also play a role in the subclinical inflammation seen in obesity.

This subclinical inflammation involves intricate cellular pathways, in conjunction with macrophage infiltration and ER stress. These pathways can amplify the activity of one another synergistically, promoting a chronic state of low-grade inflammation. In addition, inflammageing and subclinical inflammation, particularly in the context of obesity, share mechanisms such as the TLR pathways, NF-κB pathway, NLRP3 inflammasome activation, oxidative stress, mitochondrial dysfunction, immune cell population alterations, and gut microbiome changes.

Vitamin D, which is recognized for its immunomodulatory and anti-inflammatory properties, may target specific pathways connected to adipose cell senescence, inflammageing, and subclinical inflammation. These commonalities represent targets for therapeutic interventions for multiple age- and obesity-related health issues.

#### 3.3.1. Toll-like Receptors (TLRs)

TLRs are a family of proteins that play a critical role within the innate immune system. In the context of obesity, TLR2 and TLR4 act as significant mediators that connect excessive food intake and the occurrence of inflammation. Vitamin D exerts regulatory effects on immune cell responses to TLR activation, hence having an influence on the inflammatory response induced by various ligands, including fatty acids [83].

High-fat diet consumption affects the intestinal microbiome composition, increasing gut permeability to lipopolysaccharides (LPSs), which can enter the bloodstream and stimulate TLRs. This condition, also known as “metabolic endotoxaemia”, can induce systemic inflammation in individuals with obesity [84]. Furthermore, saturated fatty acids have the ability to activate TLR4 in AT, hence activating inflammatory signaling pathways [85,86] and contributing to local and systemic inflammation. Obese db/db mice, which exhibit leptin receptor dysfunction, show increased expression of TLR4 messenger RNA (mRNA) in AT. C3H/HeJ mice, which express nonfunctional mutant TLR4, show decreased adiposity and increased insulin signaling upon consumption of a high-fat diet in comparison to control animals. Silencing TLR4 increases insulin resistance and reduces inflammation in murine models of diet-induced obesity. Additionally, the same TLR4 mutation results in a reduction in macrophage proliferation. The VAT of obese mice exhibits increased expression of TLR genes compared to subcutaneous AT (SAT) [87]. Mouse studies have revealed that nonfunctional mutant TLR4 leads to reduced obesity-related symptoms and inflammation, suggesting that TLR4 plays a pivotal role in diet-induced obesity and metabolic disturbances.

Data from in vitro studies conducted by Youssef-Elabd et al. [88] and Vitseva et al. [89] showed that the exposure of human SAT explants and adipocytes to saturated fatty acids resulted in the activation of TLR4 signaling, which led to an increase in NF-κB activity. Moreover, the upregulation of TLR2 and TLR4 on macrophages and adipocytes was caused by the activation of these receptors with agonists (Pam3CSK4 (Pam3CysSerLys4) and LPSs, respectively), which increased the production of several inflammatory mediators. These results provide comprehensive evidence of the potential involvement of TLRs in the inflammatory response induced by obesity. Nevertheless, one of the limitations of this research was the lack of a control group of patients with normal weight, which would have allowed a comparison of TLR4 expression according to BMI. Catalán et al. [90] investigated the link between BMI and TLR4 expression in a recent study and found a significant increase in the expression of TLR4 mRNA in the VAT of obese persons compared with that of lean individuals. However, a difference in TLR4 mRNA expression between the two groups was not found in the SAT.

Chronic inflammation is a well-established catalyst of cellular senescence. The persistent activation of TLRs in AT might indirectly facilitate the onset of adipocyte senescence by sustaining the inflammatory response. The activation of TLRs induces the production of ROS and proinflammatory cytokines. The presence of abundant ROS has the potential to cause DNA damage and elicit various forms of cellular stress, both of which may serve as stimuli for the onset of cellular senescence [91]. Several studies have shown that TLR signaling may have a direct impact on pathways related to cellular senescence [91], such as the activation of the NF-κB pathway [92]. Furthermore, SASP has the potential to induce further stimulation of TLRs, thereby establishing a feedback loop that amplifies both ageing-related changes and inflammation [93].

Given the function of TLRs in obesity-associated inflammation and metabolic dysfunction, they are potential therapeutic targets for obesity and metabolic diseases. Inhibitors of TLR signaling might help to reduce inflammation and improve metabolic outcomes in obesity [94]. Numerous studies have proposed that vitamin D may play a role in obesity by altering TLR activity and thus influencing AT inflammation and senescence [61,95,96,97]. Vitamin D modulates the expression of TLRs on various immune cells. For instance, 1,25(OH)_2_D_3_ can upregulate the expression of TLR2 and TLR4 on human monocytes [98,99]. In an animal study, the intervention, i.e., vitamin D and regular exercise, decreased the expression of FATP4 (fatty acid transport protein 4) and TLR4 in both AT and the liver [100]. In obesity, regulating TLR levels in AT may have an impact on the susceptibility of cells to inflammatory stimuli.

Vitamin-D-deficient subjects exhibit a significant elevation of the levels of the cytokines IFN-α (interferon-α), IL-6, and TNF-α after TLR2 stimulation. However, this reaction is mitigated with vitamin D supplementation [97].

In summary, while vitamin D can influence TLR activation and has potential implications for AT inflammation and senescence, the detailed mechanisms, especially in the context of obesity, are still under investigation. Considering the multifaceted role of vitamin D in immune modulation and cellular health, the relationship between vitamin D and TLRs remains an important research topic.

#### 3.3.2. NF-κB Pathway

The NF-κB pathway is activated by a variety of TLR ligands and proinflammatory cytokines, oxidized LDL, ROS, and stress signals (Figure 3). The activation of NF-κB leads to the upregulation of miR-34a, which suppresses the production of nicotinamide adenine dinucleotide (NAD) and decreases the levels of sirtuin 1 [101]. In both obesity and ageing, the decrease in the production of sirtuin 1 may involve the NF-κB pathway. The activation of NF-κB and IL-1α in senescent cells leads to a reciprocal induction of these factors, resulting in a positive feedback loop that triggers the synthesis of SASP components [102]. Moreover, a reduction in SIRT1 expression hinders the recruitment of macrophages to AT, whereas the upregulation of SIRT1 expression prevents macrophage migration and inflammation. SIRT1 silencing in AT promotes NF-κB activation and histone hyper-acetylation at histone H3 at lysine 9 (H3K9), which is associated with the activation of inflammatory genes [103].

Vitamin D can inhibit the NF-κB pathway, potentially suppressing the inflammatory response triggered by TLR activation in AT [104]. By impeding the phosphorylation and degradation of the IκB protein (inhibitor of kappa B, a protein that binds the NF-κB transcription factor and maintains NF-κB in an inactive state), the 1,25(OH)_2_D_3_-VDR complex hampers NF-κB translocation to the nucleus. Additionally, this complex has the ability to inhibit the activation of the p65 subunit within the NF-κB complex [105] (Figure 3).

The effect of 1,25(OH)_2_D_3_ administration on NF-κB signaling pathways was evaluated in studies using cultured cells, which yielded promising results. An investigation into the impact of high-fat diet consumption on the production of proinflammatory cytokines by stromal vascular cells (SVCs) and adipocytes in lean and obese mice revealed that in vitro treatment with 1,25(OH)_2_D_3_ significantly decreased TLR2 expression and increased the mRNA levels of IκBα in SVCs [106]. Several studies have shown that 1,25(OH)_2_D_3_ increases the levels of IκBα and thus modulates the NF-κB signaling pathway in murine 3T3-L1 adipocytes, as well as in human preadipocytes and adipocytes [107]. In vitro experiments showed that 1,25(OH)_2_D_3_ decreases IL-6, IL-1β, and MCP-1 (monocyte chemoattractant protein-1) production in SVCs but has no effect on adipocytes [106].

Vitamin D can suppress the NF-κB signaling pathway by decreasing the expression of miRs (miR-146a, miR-150, and miR-155) in adipocytes derived from both humans and mice. When fed a high-fat diet, miR-146a^−/−^ mice gained more weight and accumulated more fat than wild-type mice, in addition to developing insulin resistance, glucose intolerance, and liver steatosis. This suggests that miR-146a plays a role in maintaining glucose balance both systemically and in adipocytes and may help suppress proinflammatory pathways associated with obesity [108]. The administration of vitamin D to mice fed a high-fat diet resulted in a significant decrease in miR levels [109]. Furthermore, miR-155 exerts a diverse array of impacts on metabolic and cellular processes. It specifically targets the telomere repeat-binding factor 1 (*trf1*) gene, which is essential for telomere preservation. Its downregulation is linked to vascular ageing, since restoring miR-155 can ameliorate the manifestations of vascular ageing. Suppressing the regulatory effects of miR-155 has the potential to mitigate obesity by influencing preadipocytes and macrophages, causing the latter to adopt a brown-like fat phenotype that is more metabolically active. Additionally, the expression of miR-155 is associated with the proinflammatory cytokine TNF-α, and miR-155 has the potential to inhibit the expression of sirtuin 1, an anti-inflammatory protein that exhibits decreased levels in the AT of mice fed a high-fat diet [52].

A study on vitamin-D-deficient mice fed a high-fat diet during gestation revealed the activation of inflammatory pathways in the AT of male offspring and an increased expression of RNAs and miRs linked to inflammation. The activation of the NF-κB signaling pathway was associated with the obesity index, as well as with genomic and epigenetic profiles [110]. A recent study [111] combining animal experiments and bioinformatics analyses showed that 1,25(OH)_2_D_3_ upregulated 45 mRNAs and putative target miRs and increased the activity of 87 related pathways. Furthermore, bioinformatics analysis revealed that the genes regulated by 1,25(OH)_2_D_3_ and their target miRs converged on the classical NF-κB signaling pathway. Vitamin D supplementation was found to activate AMPK (AMP-activated protein kinase) and suppress NF-κB phosphorylation in AT in a study on male C57BL/6J mice, thus having positive effects on AT growth, macrophage recruitment, and inflammation [9].

A study on human white preadipocytes found that VDR ligands, specifically 1,25(OH)_2_D_3_, affect gene expression and the secretion of inflammatory factors upon stimulation with IL-1β. The study also found that IL-1β may promote metaflammation in AT by elevating the expression of proinflammatory genes, either by increasing RelA/p65 phosphorylation or by lowering RelA/p65 methylation within the canonical NF-κB pathway [112]. Furthermore, 1,25(OH)_2_D_3_ reduces IL-6 secretion in a dose-dependent manner in human adipocytes. It inhibits NF-κB translocation into the nucleus and reduces IL-6 mRNA levels, suggesting a potential therapeutic effect for vitamin D in inflammation [113]. Gao et al. [114] observed a notable difference in the levels of secreted IL-6, IL-8, and MCP-1 between preadipocytes and mature adipocytes. Based on this finding, it appears that preadipocytes may significantly contribute to the synthesis of proinflammatory mediators. Furthermore, 1,25(OH)_2_D_3_ significantly reduces the secretion of MCP-1, IL-6, and IL-8 by preadipocytes. Preadipocytes display a higher level of IκBα protein after 1,25(OH)_2_D_3_ administration, suggesting the involvement of NF-κB signaling in the effect of 1,25(OH)_2_D_3_ [114].

An additional investigation using cultured human adipocytes revealed that 1,25(OH)_2_D_3_ inhibits the NF-κB pathway, hence impeding macrophage activation. This was confirmed by the increase in the baseline IκBα levels and the reversal of IκBα suppression induced by a macrophage-conditioned medium. The significant decrease in the release of IL-6 and IL-8 after NF-κB/p65 inhibition indicated that NF-κB/p65 is crucial for the production of proinflammatory cytokines in human preadipocytes [115].

A case–control study examined the correlation between vitamin D insufficiency and AT inflammation among patients diagnosed with colorectal cancer. One of the noteworthy discoveries was the strengthened relationship between the transcriptional activity of VDR and the levels of inflammatory indicators, including NF-κB1, IL-6, and IL-1β, in AT. The negative association between serum 25(OH)D levels and NF-κB1 expression indicated a potential involvement of vitamin D in modulating the NF-κB inflammatory pathway. This relationship was additionally supported by the identified connections between 25(OH)D and epigenetic alterations, specifically DNA methyltransferase 3a (DNMT3A) and VDR DNA methylation. These findings potentially suggest the wider involvement of vitamin D in the epigenetic control of inflammation [116].

#### 3.3.3. NLRP3 Inflammasome Activation

NLRP3 is an intracellular complex composed of multiple proteins that is primarily expressed in macrophages. It is critical for the innate immune response, as it is responsible for identifying and reacting to specific microbial signals, cellular injury, and metabolic abnormalities [40,117]. NLRP3 inflammasome activation requires cell priming, triggered by the binding of microbial components (LPSs) and cytokines (TNF-α) to TLRs. This activates the NF-κB pathway, which induces the transcription and synthesis of NLRP3, which remains inactive. The activation of the NLRP3 inflammasome is caused by numerous factors, such as mitochondrial damage, ROS production, and lysosomal rupture [40]. After NLRP3 activation, the inflammasome is assembled through the activation of caspase-1 and the conversion of IL-1β and IL-18 into their mature, functional forms, which are then released from the cell to promote inflammation and pyroptosis [118] (Figure 4).

The inhibition of NLRP3 inflammasome activation by vitamin D may reduce the secretion of IL-1 and IL-18. Vitamin D acts as a regulator of NLRP3 inflammasome activation by modulating the activity of macrophages and dendritic cells. In a study on mice with obesity and asthma, vitamin D_3_ levels were related to the increased expression of IL-1β mRNA and NLRP3 mRNA in lung tissue [119].

Vitamin D can influence cellular ROS levels, and this could be one of the mechanisms through which it modulates NLRP3 inflammasome activity [120]. ROS and cellular oxidative stress may be induced by vitamin D metabolites due to increased lipid oxidation; therefore, 1,25(OH)_2_D_3_ was postulated to be a “priming” molecule that can indirectly induce NLRP3 inflammasome activation to promote IL-1β maturation [120]. Given the involvement of the NF-κB pathway in the priming phase of NLRP3 inflammasome activation, the modulation of the NLRP3 inflammasome can be facilitated by vitamin D through inhibition of the NF-κB pathway. Moreover, vitamin D has been reported to induce autophagy in some cells, and through this mechanism, it might influence NLRP3 inflammasome activity [40].

The exact role of vitamin D in NLRP3 inflammasome activation and the underlying mechanism are not fully understood and need to be explored. There is little evidence from experimental and clinical studies of the beneficial and harmful effects of vitamin D on the inflammatory process triggered by NLPR3 inflammasome activation. Given the importance of the NLRP3 inflammasome in metabolic diseases, assessing the relationship between vitamin D and this inflammatory complex may have therapeutic implications.

#### 3.3.4. Immune Cell Modulation

Inflammation can be reduced by vitamin D, which influences the activity of dendritic cells, macrophages, and T cells and affects both the innate and adaptive immune systems [12]. 1,25(OH)_2_D_3_ is essential for the downregulation of numerous cytokines in preadipocytes and adipocytes, including MCP-1, IL-1β, IL-6, and IL-8 [106,121,122]. The presence of VDR in both adipocytes and macrophages suggests the crucial roles of vitamin D and its metabolites in inflammatory processes within the AT [123,124].

##### The Anti-Inflammatory Action of Vitamin D in Cultured Cells

The anti-inflammatory properties of vitamin D were demonstrated in research involving the incubation of adipocytes with 1,25(OH)_2_D_3_ for either 24 or 48 h before inflammatory stimuli were introduced [57,66,114,115]. In response to 1,25(OH)_2_D_3_, the mRNA levels of proinflammatory cytokines (IL-6, IL-8, and CD14) decreased in human adipocytes and 3T3-L1 preadipocytes [121,122]. The possible mechanisms by which vitamin D exerts its anti-inflammatory effect include a decreased protein expression of TLR-2 and TLR-4, an increased mRNA expression of trans-acting T-cell-specific transcription factor (GATA-3) via the upregulation of the upstream factor signal transducer and the activator of transcription 6 (STAT6), decreased levels of pp38 (phosphoprotein 38) and p42/44 (ERK1/2), and an altered localization of p65 [125].

In an observational study on vitamin-D-deficient women with obesity, macrophage infiltration of AT was dose-dependently associated with serum vitamin D levels. Participants with moderate vitamin D deficiency exhibited a greater concentration of proinflammatory cytokines than those with mild deficiency [126]. Supplementation with vitamin D_3_ decreased the production of proinflammatory cytokines: TNFα, PAI-1 (plasminogen activator inhibitor-1), IL-6, and ROS and pro-fibrotic genes in studies on AT biopsies from patients with obesity and 25(OH)D insufficiency [127]. Reductions in TNF α and IL-6 levels were also observed in a group of patients with severe obesity undergoing bariatric surgery [128].

Incubation of adipocytes from postmenopausal women with obesity with 1,25(OH)_2_D_3_ decreases the IL-1β-induced secretion of IL-8 protein but has no significant effect on the production of IL-6 and MCP-1 [129]. The pretreatment of differentiated adipocytes with 1,25(OH)_2_D_3_ followed by incubation with TNFα or concurrent incubation with TNFα and 1,25(OH)_2_D_3_ decreases the mRNA expression of MCP-1 [130].

Studies on the anti-inflammatory effect of vitamin D on AT have provided encouraging results, suggesting its beneficial effect on local low-grade inflammation associated with obesity. Furthermore, these studies offer detailed insight into the local mechanisms triggered by vitamin D in AT.

##### Inconsistent Results in Human Studies

Research examining the effect of vitamin D supplementation on serum levels of inflammatory cytokines in humans has yielded inconclusive findings [131,132,133]. Vitamin D supplementation increases the levels of anti-inflammatory cytokines (IFN-γ and IL-10) in subjects with normal weight and vitamin D insufficiency [134]. While some studies have suggested that increased serum 1,25(OH)_2_D_3_ concentrations are associated with decreased inflammatory marker expression in individuals with normal weight [135], other research has found no significant effects of vitamin D on inflammatory biomarkers in patients with obesity [136,137,138,139,140,141,142,143]. Furthermore, studies on the effect of vitamin D supplementation on CRP serum levels in patients with obesity have yielded conflicting results [137,138,139,141,142,143,144,145,146,147,148,149,150,151,152,153,154]. A key factor contributing to this inconsistency is the variation in study design across different trials. First, the sample population of the RCTs ranged from individuals with normal weight to those with obesity, as well as from patients with T2DM and those without T2DM. Importantly, the response to vitamin D supplementation may differ among these groups, as underlying health conditions and metabolic status can influence cytokine production and regulation. Furthermore, the duration of the studies also varied, ranging from weeks to months. It is well established that cytokine levels can fluctuate over time, and the duration of supplementation may influence the observed effects. Therefore, differences in study duration among different RCTs may have contributed to the inconsistent findings. Another significant discrepancy among the RCTs was the dose or form of vitamin D used for supplementation. In some studies, vitamin D was administered orally; however, in others, vitamin D was given via parenteral routes. Additionally, the dosage of vitamin D varied across trials. It is essential to consider the impact of the route of administration and dosage on the bioavailability and absorption of vitamin D, which in turn may influence its effectiveness in modulating cytokine levels.

##### Insights from Meta-Analyses

A meta-analysis conducted by Yu et al. [155] showed that vitamin D supplementation reduced CRP levels but did not impact IL-6 or TNF-α levels. The analyzed studies varied in terms of design, vitamin D dose, and duration. Another meta-analysis by Jamka et al. [156] did not find a significant effect of vitamin D supplementation on inflammatory marker levels in overweight and obese patients. Few studies were included in this analysis, and they used different vitamin D doses and treatment durations, making it challenging to identify clear patterns of changes [8]. A recent umbrella meta-analysis of 23 studies found a significant change in serum CRP and TNF-α levels after vitamin D supplementation. However, the authors reported a high degree of heterogeneity among the studies, which might have been attributed to the fact that the studies included patients with different diseases (T2DM, chronic kidney disease, inflammatory bowel disease, and heart failure) [8]. Thus, the current research findings underscore the need for more extensive and comprehensive studies to better understand the influence of vitamin D supplementation on inflammatory cytokine levels in humans [61].

#### 3.3.5. Oxidative Stress

Excessive food intake results in a positive energy imbalance, which favors oxidative stress in adipose cells through NADH (nicotinamide adenine dinucleotide hydrogenase) overproduction and the electron transport chain in mitochondria. Excessive ROS production through NADPH oxidase 4 (NOX4) is linked to oxidative stress in adipocytes [157,158,159,160]. Oxidative stress amplifies local inflammation and is accompanied by the release of proinflammatory cytokines [161]. Exacerbated oxidative stress and local hypoxia due to the imbalance between the supply of and demand for hypertrophic adipose cells together with the release of proinflammatory molecules provoke adipocyte damage [159] (Figure 5).

The antioxidant activity of vitamin D is linked to its ability to increase the synthesis of superoxide dismutase (SOD) and glutathione peroxidase (GPx) in AT [162] and to limit NOX and ROS production in adipocytes exposed to high glucose levels [22]. Vitamin D has been found to mitigate detrimental oxidative stress caused by hyperleptinemia in human endothelial cells, suppress the proinflammatory effects of leptin, and increase intrinsic antioxidant mechanisms in the cell through the upregulation of many antioxidant genes. The observed protective effects of 1,25(OH)_2_D_3_ appear to be contingent upon the presence of VDR [163].

AT senescence is strongly linked to oxidative stress via a multitude of interconnected pathways. Excess ROS can cause cellular damage by promoting the release of SASP components, specifically proinflammatory cytokines, in adipocytes [39]. The accumulation of senescent cells amplifies oxidative stress, hence creating a feed-back loop that further hinders the function of AT [164]. The activation of the Nrf2 signaling pathway, which serves as a master regulator of the antioxidant response, has the potential to mitigate oxidative damage (Figure 5). Nevertheless, chronic stress might overcome this defense mechanism, resulting in the disruption of adipogenesis and insulin signaling [27]. Furthermore, adipocyte senescence is facilitated by the overactivation of the p53/p21RAS and p16INK4a signaling pathways, which further disrupts the homeostasis of AT and metabolic activities [102]. Oxidative stress plays a pathophysiological role in the senescence of AT and the progression of metabolic disorders through this complex network.

The administration of 1,25(OH)_2_D_3_ to 3T3-L1 adipocyte cells exposed to high glucose levels appeared to lower the production of ROS. A decrease in NOX4 expression together with increased expression of the antioxidants Nrf2 and Trx (thioredoxin) accompanied the antioxidant effect of 1,25(OH)_2_D_3_. These findings highlight the capacity of 1,25(OH)_2_D_3_ to mitigate oxidative stress through the modulation of the NOX4/Nrf2/Trx-signaling pathway in the presence of elevated glucose levels [22].

In cells from rats fed a high-fat diet, vitamin D treatment for 5 weeks increased antioxidant enzyme (GPx and SOD) levels [162]. Furthermore, a significant reduction in oxidative stress levels was observed when VAT samples obtained from individuals who underwent abdominal surgery were incubated with 1,25(OH)_2_D_3_ compared to those that were not incubated with vitamin D [165]. A positive relationship was observed between vitamin D deficiency, oxidative stress, and VAT accumulation in the participants with obesity [166]. Another study involving patients with T2DM and obesity revealed that more severe obesity was correlated with increased oxidative stress (reflected by serum H_2_O_2_ concentrations) and diminished serum 1,25(OH)_2_D_3_ concentrations compared to their less obese or overweight T2DM counterparts [167]. Furthermore, a study including individuals with metabolic syndrome and T2DM who were given 1,25(OH)_2_D_3_ as a supplement revealed that vitamin D effectively reduced oxidative DNA damage and increased insulin sensitivity, as seen by a decline in the HOMA-IR (homeostatic model assessment for insulin resistance) and TG/HDL ratio (triglyceride (TG) to high-density lipoprotein (HDL) [168]. Additionally, vitamin D supplementation decreased the extent of oxidative DNA damage in the lymphocytes of patients aged 45 and older with metabolic disorders and vitamin D deficiency [168].

The literature presents inconsistent findings pertaining to the potential pro- and antioxidant effects of vitamin D on AT, signifying the need for continued exploration. An increase in ROS generation and impairment of the mitochondrial membrane potential were observed in adipocytes exposed to elevated glucose or free fatty acid concentrations [30]. Exposing 3T3-L1 adipocytes to an elevated concentration of glucose followed by the addition of 1,25(OH)_2_D_3_ resulted in a significant increase in ROS production and NADPH oxidase activity. This increased ROS generation could be attributed to factors such as mitochondrial uncoupling inhibition and intracellular calcium ion flow in adipocytes, especially under hyperglycemic conditions [121].

Given the pivotal role of mitochondria in ROS generation, elucidating the effects of active vitamin D forms on ROS generation and mitochondrial dysfunction under hyperglycemic conditions is crucial. The existing literature offers mixed perspectives on vitamin D’s pro- and antioxidant effects, emphasizing the need for continuous research. The primary consensus from experimental research is that vitamin D’s antioxidant effects largely stem from its ability to modulate the expression of genes associated with antioxidant defense mechanisms. Nevertheless, considering the limited evidence regarding the appropriate dosage of vitamin D and the duration of vitamin D treatment, it is crucial to conduct a comprehensive and thorough investigation, not only involving measurement of serum vitamin D levels but also an analysis of AT biopsies [169]. In addition, meta-analyses are needed to pinpoint the exact ramifications of vitamin D administration on oxidative status within AT.

## 4. Conclusions

Subclinical inflammation and inflammageing share common pathways that underlie the development of metabolic and cardiovascular complications associated with obesity. It is not well known whether inflammageing is an additional risk factor for local AT inflammation associated with obesity or whether local inflammation is a trigger that exacerbates inflammageing in AT. The additional effects of these two mechanisms are yet to be determined. Further studies on the role of inflammation through specific markers, such as SASP components and SA β-gal, are needed. Furthermore, studies on local AT inflammation in patients with obesity and patients with metabolic syndrome would provide more detail regarding the differences between these diseases.

Differences in the design of RCTs investigating the effects of vitamin D on inflammatory cytokines contribute to the inconsistency of the results. Serum levels of cytokines are commonly used as biomarkers to evaluate the systemic inflammatory status of individuals. By measuring the levels of various cytokines, researchers can assess the overall inflammatory response in the body. VDR, which is present in numerous tissues, including immune cells, plays a pivotal role in modulating the immune response and inflammation. Importantly, changes in serum cytokine levels may not necessarily reflect the exact alterations in the AT. Given the widespread distribution of VDR, these changes in cytokine levels can be interpreted as indicating a general anti-inflammatory response. These changes directly suggest potential concurrent local changes in AT. They also provide valuable insights into the overall anti-inflammatory influence of vitamin D and its potential effects on AT. As AT is a major site for cytokine production and immune cell infiltration, it is reasonable to hypothesize that the observed changes in systemic cytokine levels are indicative of parallel modifications within the AT microenvironment.

Further research is necessary to fully elucidate the precise mechanisms through which vitamin D and the systemic anti-inflammatory response impact local AT inflammation. Utilizing techniques such as AT biopsies or employing imaging modalities to assess AT composition can provide more specific and direct measurements of local changes. These complementary approaches will enhance our understanding of the intricate interplay between vitamin D, systemic inflammation, and AT inflammation.

Furthermore, obesity and senescence of adipose cells are distinct processes that share common pathways that promote subclinical chronic inflammation, but whether their effects are additive or exponential remains to be further studied. The physiological effects of vitamin D at the molecular and genomic levels in obesity can be studied to further investigate the process of inflammation in AT.

In conclusion, the evaluation of serum cytokine levels in RCTs provides an indirect assessment of the anti-inflammatory response associated with VDR-targeted interventions. While changes in serum cytokine levels serve as a valuable indicator of systemic changes, it is important to investigate cytokines levels in AT to gain a comprehensive understanding of their impact on AT inflammation. By combining findings from both systemic analysis and studies of AT, we can improve our knowledge regarding the overall therapeutic potential of vitamin-D-related interventions in modulating inflammation.

## Figures and Tables

**Figure 1 metabolites-14-00004-f001:**
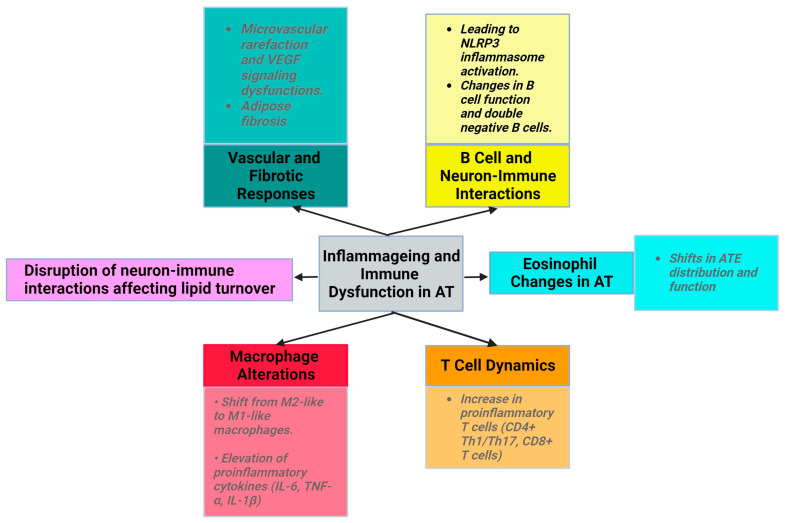
Characteristics of senescence in AT (created with Biorender.com (accessed on 11 December 2023)).

**Figure 2 metabolites-14-00004-f002:**
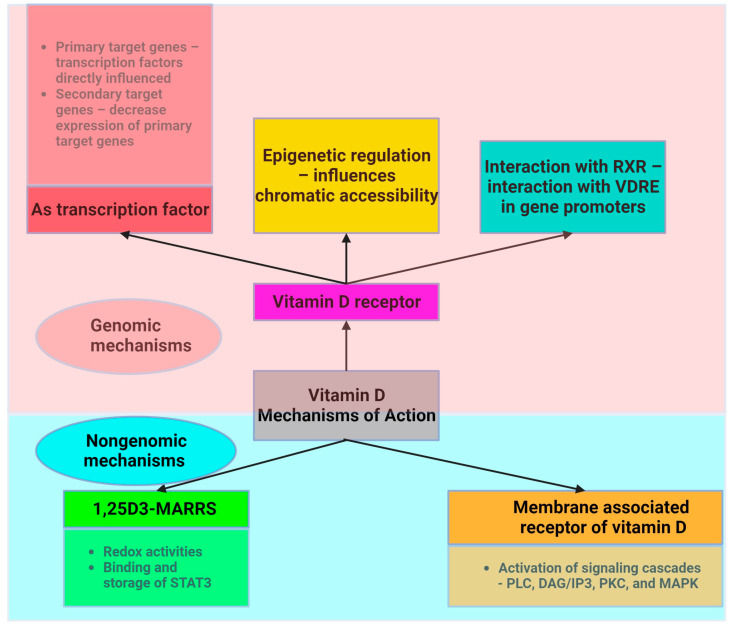
Intracellular mechanism of action of vitamin D (created with Biorender.com).

**Figure 3 metabolites-14-00004-f003:**
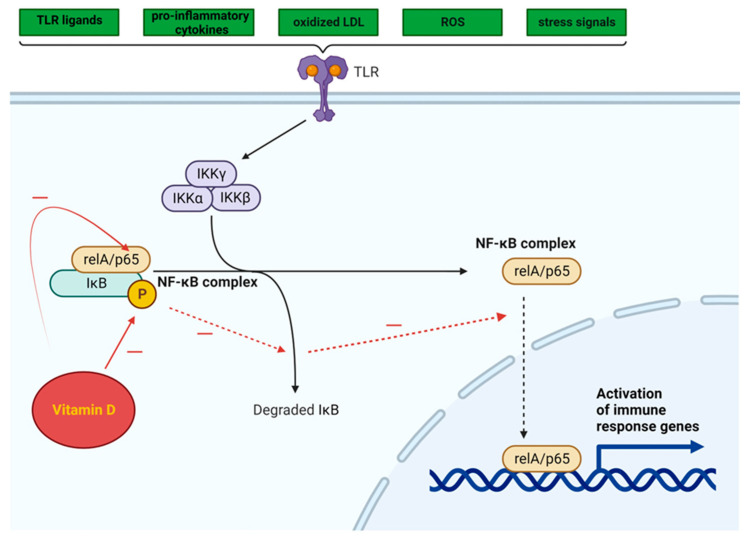
Intracellular NF-κB pathway and influence of vitamin D (adapted from “NF-KB signaling pathway”, by Biorender.com (2023). Retrieved from https://app.biorender.com/biorender-templates, accessed on 12 November 2023).

**Figure 4 metabolites-14-00004-f004:**
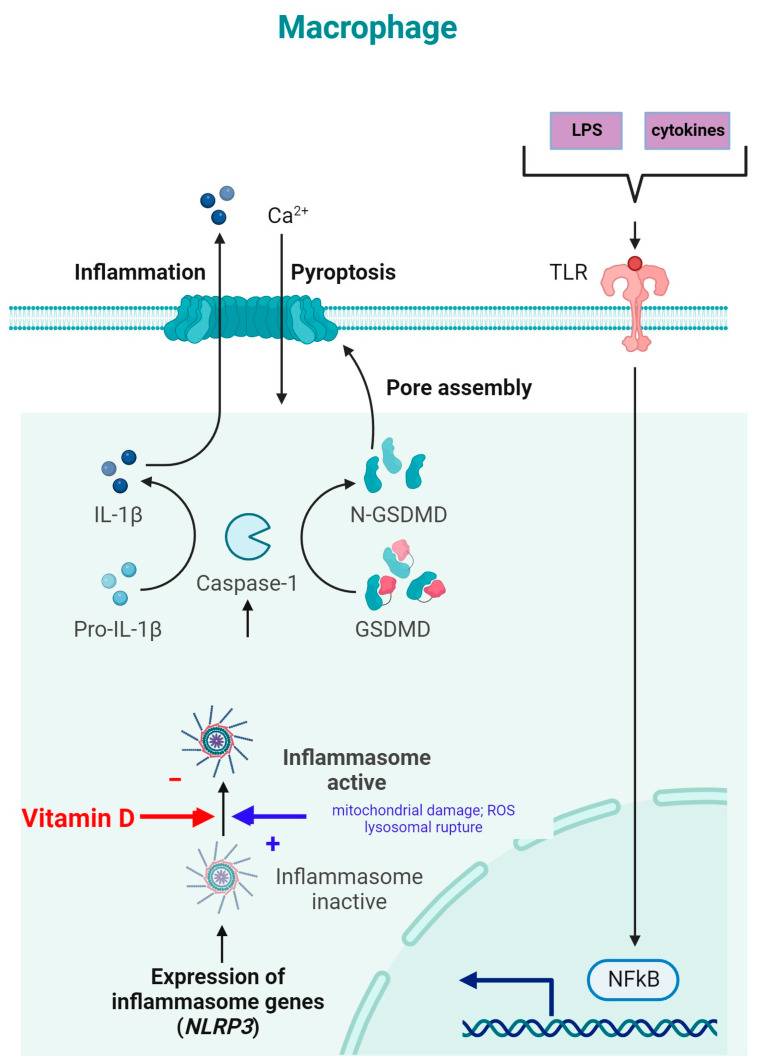
NLRP3 inflammasome modulation by vitamin D (adapted from “Suppression of Inflammasome by IRF4 and IRF8 is Critical for T Cell Priming”, by Biorender.com (11 December 2023). Retrieved from https://app.biorender.com/biorender-templates (accessed on 11 December 2023)).

**Figure 5 metabolites-14-00004-f005:**
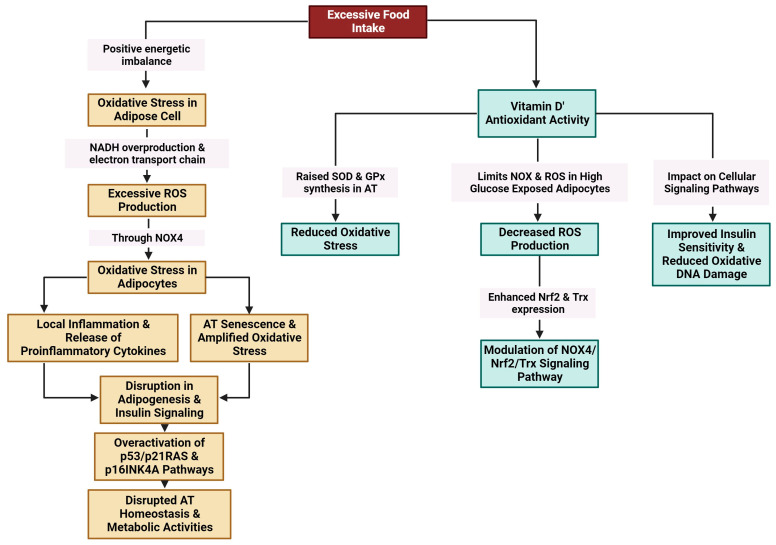
Modulation of oxidative stress in AT by vitamin D (created with Biorender.com (accessed on 11 December 2023)).

**Table 1 metabolites-14-00004-t001:** Search terms and phrases.

No.	Concept	Search Terms
	PUBMED	
**1**	**Vitamin D**	(‘vitamin D’ [MeSH Terms] OR ‘ergocalciferols’ [MeSH Terms] OR ‘vitamin D’ [All Fields] OR ‘ergocalciferols’ [MeSH Terms] OR ‘ergocalciferols’ [All Fields]) OR ‘treatmen’ [Title/Abstract]) OR ‘supplementation’ [Title/Abstract] OR ‘vitamin D3’ [Title/Abstract] OR ‘vitamin D2’ [Title/Abstract])
**2**	**Weight**	(‘obesity’ OR ‘overweight’ OR ‘fat’ OR ‘obese’ OR ‘unhealthy weight’ OR ‘high BMI’)
**3**	**Senescence**	(‘senescence’ OR ‘senescent cells’ OR “aging” [MeSH Terms] OR ‘aging’ [All Fields] OR ‘senescence’ [All Fields] OR ‘senesce’ [All Fields] OR ‘senesced’ [All Fields] OR ‘senescences’ [All Fields] OR ‘senescent’ [All Fields] OR ‘senescents’ [All Fields] OR ‘senesces’ [All Fields] OR ‘senescing’ [All Fields])
**3**	**TLR**	(‘Toll-Like Receptors’ OR ‘TLR’)
**4**	**NLRP3 inflammasome**	(‘NLRP3 Inflammasome’) OR ‘inflammasome’
**5**	**Oxidative stress**	(‘oxidative stress’ OR ‘reactive oxygen species’ OR ‘ROS’ OR ‘oxidative stress’ [MeSH Terms] OR (‘oxidative’ [All Fields] AND ‘stress’ [All Fields]) OR ‘oxidative stress’ [All Fields])
**6**	**Inflammatory markers**	(‘inflammatory markers’ OR ‘cytokines’ OR ‘interleukins’ OR ’CRP’ OR ‘C-reactive protein’ OR ‘tumor necrosis factor’ OR ‘TNFα’ OR ‘adipokines’ [MeSH Terms] OR ‘adipokines’ [All Fields] OR ‘adipokine’ [All Fields])
**7**	**NF-κB pathway**	(‘NF-kappaB signaling pathway’ OR ‘NF-κB pathway’)
**8**	**CRP**	(‘high sensitivity C-reactive protein’ [MeSH Terms]) OR ‘high-sensitivity C-reactive protein’ [MeSH Terms]) OR ‘C-reactive protein’ [MeSH Terms]) OR ‘high-sensitive C-reactive protein’ [MeSH Terms]) OR ‘high sensitive C-reactive protein’ [MeSH Terms]) OR ‘CRP’ [Title/Abstract]) OR ‘hsCRP’ [Title/Abstract])
**3**	**Combinations**	1 AND 2 AND 3; 1 AND 2 AND 4; 1 AND 2 AND 5; 1 AND 2 AND 5; 1 AND 2 AND 6; 1 AND 2 AND 7
	**EMBASE/EBSCO**	
**1**	**Adipose tissue**	(‘adipocyte inflammation’ OR ‘adipose tissue inflammation’ OR ‘interleukin’ OR ‘inflammasomes’ OR ‘adipos * inflammageing’)
**2**	**Vitamin D**	(‘vitamin D’ OR ‘free vitamin D’ OR ‘vitamin D action’ OR ‘vitamin D metabolism’ OR ‘classical action of vitamin D’ OR ‘non-classical action of vitamin D’ OR ‘genomic action of vitamin D’ OR ‘non-genomic action of vitamin D’ OR ‘vitamin D receptor’ OR ‘vitamin D deficiency’ OR ‘vitamin D’ OR ‘vitamin D supplementation’)
**3**	**Obesity**	(‘obesity’ OR ‘overweight’ OR ‘body mass index’ OR ‘fat mass’ OR ‘abdominal obesity’ OR ‘visceral adipose tissue’ OR ‘subcutaneous adipose tissue’ OR ‘human adipose tissue’ OR ‘human preadipocyte’ OR ‘human adipocyte’ OR ‘adipocyte differentiation’ OR ‘adipogenesis’ OR ‘adipose tissue function’ OR ‘weight loss’ OR ‘bariatric surgery’ OR ‘gastric bypass surgery’)
**4**	**Inflammation**	(‘adipocyte inflammation’ OR ‘adipose tissue inflammation’ OR ‘interleukin’ OR ‘inflammasomes’ OR ‘inflammation’ OR ‘inflammatory cytokines’ OR ‘adipokines’ OR ‘cytokine’ OR ‘immune response’ OR ‘leptin’ OR ‘adiponectin’ OR ‘TNF-α’ OR ‘C-reactive protein’ OR ‘interleukin’ OR ‘insulin secretion’ OR ‘insulin resistance’ OR ‘glucose homeostasis’ OR ‘HOMA-B’ OR ‘HOMA-IR’)
**5**	**Senescence**	(‘adipocyte senescence’ OR ‘adipose tissue senescence’ OR ‘senescence’ OR ‘senescent’ OR ‘adipose inflammaging’)
	**Combinations**	(1 OR 2 OR 3) AND 4; (1 OR 2 OR 3) AND 5

* shows the root of the keyword.

**Table 2 metabolites-14-00004-t002:** Comparison of the changes associated with senescence and obesity in AT.

Changes Associated with Senescence	Changes Associated with Obesity
Accumulation of senescent cells in AT [19].	Increases in fat mass and the number of large, hypertrophic adipocytes [7].
Elevated SA β-Gal activity indicative of cellular ageing [24].	Redistribution of fat from subcutaneous to visceral depots, contributing to metabolic dysfunction [2,26].
Presence of senescent cells exhibiting a SASP [24].	Increased inflammation and infiltration of immune cells [7].
Increased secretion of inflammatory and profibrotic factors [33].	Dysfunction of endothelial cells leading to vascular challenges and inflammation [35].
Impaired preadipocyte differentiation and functionality [24].	Hypoxia due to insufficient vascularization [36].
Increased immune cell infiltration driven by factors secreted by senescent cells [1].	Insulin resistance and risk of type 2 diabetes mellitus (T2DM) [25].
Shift in AT distribution and function [26,38].	Obesity-related lipotoxicity and metabolic imbalances [39].

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
