# Peer review of "From the Sun to the Cell: Examining Obesity through the Lens of Vitamin D and Inflammation"

_metabolites, 2023, doi:10.3390/metabo14010004_

Round 1

Reviewer 1 Report

Comments and Suggestions for Authors

Major comments:

1. The figures reflect only a minor proportion of the text. Please provide more figures and rather reduce the amount of text, i.e. please be as visual as possible and focus on the essential messages. Moreover, please improve the readability of the text in the figures via higher font size, more contrasting color etc.

2. The manuscript needs to be harmonized in style of text, abbreviations and figures. Apparently, different parts of the manuscript were written by different authors without coordinations. Some parts of the manuscript are even repetitive. 

Minor comments:

1. Please define all abbreviations at their first time use and apply it them then consistently (many terms are defined more than once others, such as NF-kB even in alternative ways ). This applies also to the Abstract.

2. Please check the whole manuscript for typos, missing points and commas as well as for unnecessary gaps.

3. When using the term "vitamin D", please be more specific, which metabolite is meant. Moreover, please do not use different names for the same compound (e.g. 1,25(OH)2D3 and calcitriol).

4. Please check all gene and protein names for the latest nomenclature and correct.

Comments on the Quality of English Language

Many typos and inconsistencies.

Author Response

Dear reviewer,

Reviewer 2 Report

Comments and Suggestions for Authors

Here are my comments and suggestions to improve the quality of your manuscript:

1-Lines 55-57, The authors highlight the concept of AT hyperplasia, but it is well-known that the number of adipocytes is determined until 4 years old, and after that, these numbers don't change.

So the hyperplasia of AT is not possible in adults, whether the hypertrophy of AT is the major cause of increased adiposity.

Specify this point and increase this part in your sentences.

2-Lines 78-79 "Several Studies" could substituted by an Umbrella meta-analysis that it is more correct because reference 8 is an Umbrella review.

3-Lines 131-132, Please better detail Activin A's functions and role.

4-Figure 1, Improve the quality of the text in the figure 1. It isn't easy to read the text.

5-Table 2, In the Legenda please cite the paper's source of your information.

6-Lines 279-280 Provide a detailed list of the local cells that synthesise Vitamin D in the active form.

7-Figure 2, this figure is too small and is difficult to read. Improve the quality of the figure magnifying it.

8-Lines 355-357 The authors consider optimal levels of 25(OH)D in the serum range between 30 and 50 ng/mL, but many national and international guidelines recommend a level between 30 to 100 ng/mL. So, the authors could explain this choice of this strict range.

9-line 426 the acronym TLR and NF-kB were explicitated in the previous lines so the full term is not necessary.

10-Line 607 and 644 " the text continues here" Is it a typo?

Comments on the Quality of English Language

None

Author Response

Dear Reviewer,

Round 2

Reviewer 1 Report

Comments and Suggestions for Authors

none

Comments on the Quality of English Language

none